# Density Functional Theory Study of CuAg Bimetal Electrocatalyst for CO$_2$RR to Produce CH$_3$OH

Sensen Xue [1], Xingyou Liang [1], Qing Zhang [1], Xuefeng Ren [2], Liguo Gao [1], Tingli Ma [3,4] and Anmin Liu [1,*]

1 State Key Laboratory of Fine Chemicals, School of Chemical Engineering, Dalian University of Technology, Dalian 116081, China; xuess@mail.dlut.edu.cn (S.X.)
2 School of Ocean Science and Technology, Dalian University of Technology, Panjin 124221, China
3 Department of Materials Science and Engineering, China Jiliang University, Hangzhou 310018, China; tinglima@life.kyutech.ac.jp
4 Graduate School of Life Science and Systems Engineering, Kyushu Institute of Technology, 2-4 Hibikino, Wakamatsu, Kitakyushu 808-0196, Fukuoka, Japan
* Correspondence: liuanmin@dlut.edu.cn

**Abstract:** Converting superfluous CO$_2$ into value-added chemicals is regarded as a practical approach for alleviating the global warming problem. Powered by renewable electricity, CO$_2$ reduction reactions (CO$_2$RR) have attracted intense interest owing to their favorable efficiency. Metal catalysts exhibit high catalytic efficiency for CO$_2$ reduction. However, the reaction mechanisms have yet to be investigated. In this study, CO$_2$RR to CH$_3$OH catalyzed by CuAg bimetal is theoretically investigated. The configurations and stability of the catalysts and the reaction pathway are studied. The results unveil the mechanisms of the catalysis process and prove the feasibility of CuAg clusters as efficient CO$_2$RR catalysts, serving as guidance for further experimental exploration. This study provides guidance and a reference for future work in the design of mixed-metal catalysts with high CO$_2$RR performance.

**Keywords:** CuAg bimetal; CO$_2$RR; pathway; theoretical study





## 1. Introduction

The increasing emissions of greenhouse gases such as CO$_2$ and CH$_4$ into the air fuels the greenhouse effect and the concomitant global warming. The increase of the concentration of CO$_2$ has the greatest impact on rising global temperatures. Owing to its important environmental and economic repercussions, reducing carbon emissions without affecting social and economic development has become one of the most urgent problems in China.

Before now, numerous strategies have been developed to reduce the concentration of CO$_2$ in the atmosphere, such as CO$_2$ capture and storage [1], absorption [2], and chemical conversion [3–6]. In recent years, the conversion of CO$_2$ into a variety of economically valuable chemicals has emerged as a promising technology [7,8]. In this regard, electrochemical CO$_2$ reduction (CO$_2$RR) stands out because of its high value-added products (including ethanol and methanol) and its favorable conversion efficiency [9–13]. Among the various catalysts for CO$_2$RR, copper (Cu) is widely used because of its abundance and ambient-environment working conditions; however, its limited activity and selectivity has hindered further development [14–18]. Consequently, recent research efforts have been devoted to enhancing its catalytic performance by adjusting catalyst properties such as particle size [19], shape [20–22], and grain boundary [23].

As a result, more attractive Cu-based bimetallic CO$_2$RR catalysts with enhanced activity and selectivity have been discovered [24–28]. Thus, Kim et al. [29] reported a CuAu catalyst demonstrating high CO selectivity (~80%) in an overpotential of ~200 mV. Zhang et al. [30] fabricated a CuPd nanoalloy with a twofold enhancement in Faradaic efficiency for CO$_2$RR to methane compared with Cu alone. Sarfraz et al. [31] reported a stable CuSn catalyst exhibiting a Faradaic efficiency for CO$_2$RR to CO greater than 90%

and a current density of $-1.0$ mA cm$^{-2}$ at $-0.6$ V vs. a reversible hydrogen electrode (RHE). Chungseok et al. [32] reported on the performance of a CuAg nanowire. The Cu-Ag interface of the CuAg nanowire significantly improved the selectivity of $CO_2$ reduction to $CH_4$, with a maximum Faraday efficiency (FE) of 72% production at $-1.17$ V (relative to an RHE). Nevertheless, studies on Cu bimetallic catalysts inducing $CO_2$RR to alcohol are scarce, and a more fundamental understanding is required for the rational design of Cu-based catalysts in this field.

After much research, significant progress has been made in $CO_2$RR, especially in experimental studies. However, traditional experimental methods have inherent limitations, including blindness, workload, time, and resource waste. Nowadays, quantum chemistry and molecular dynamic simulations are widely used for calculating and simulating chemical systems as a technique to understand and predict behaviors at the molecular level [33–37]. Combining computational chemistry predictions with experimental investigation is a highly efficient methodology for accelerating the investigation of $CO_2$RR. In electrocatalysis research, the density functional theory (DFT) is widely used to analyze possible reaction path energy changes, reaction mechanisms, and catalyst materials for special reactions, such as $CO_2$RR, NRR, ORR, and so on [38]. $CH_3$OH is one of the most important chemicals used as a green fuel or the intermediate of reactions. It can be directly used in energy conversion systems such as methanol fuel cells or internal combustion engines, due to its relatively high energy density.

In this study, we theoretically designed a series of CuAg bimetallic clusters and investigated their feasibility as catalysts for $CO_2$RR to methanol according to the DFT. Considering that Ag nanoparticles have been previously shown to act as nucleation seeds for CuAg particles [39], we selected the most stable Ag cluster as a basis for the design of CuAg clusters with different configurations. Then, the $CO_2$ adsorption behavior on different sites was evaluated. Finally, the reaction pathway was fully studied to determine the optimal CuAg cluster configuration. This work sought to experimentally study a series of CuAg bimetallic cluster catalysts and establish a screening mechanism to calculate the catalytic activity of reducing $CO_2$RR to $CH_3$OH. This work can serve as guidance for further experimental exploration.

## 2. Results and Discussion

### 2.1. Study on the Stability of the Ag Clusters

In contrast with $Ag_1$ and $Ag_2$ clusters, which exhibit only one structure, different isomers appear when the number of Ag atoms (n) is n ≥ 3. In this case, the most stable configuration among these isomers must be screened. For n = 3, there are two isomers, i.e., linear and triangular, with similar bond lengths according to the calculation. By comparing their total energy and binding energy (Table S1), it was found that the binding energy of the triangular isomer is lower than that of its linear counterpart. Therefore, the most stable structure of the $Ag_3$ cluster is a flat regular triangle; its optimized structure is shown in Figure S1.

When n = 4, $Ag_4$ exhibits five isomeric structures (Figure S2), among which the most stable is a flat rhombus. For n = 5, the most stable configuration is a plane isosceles trapezoid composed of three triangles. Moreover, when n = 6, a two-dimensional close-packed shape composed of four triangles has the lowest binding energy and is therefore the most stable. Surprisingly, these selected structures are all planar (the corresponding calculation results can be found in Supplementary Information).

Figure 1 depicts the most stable $Ag_n$ cluster structures for n = 1–6. The relationship between the stability of the six clusters and the number of Ag atoms is analyzed in Figure 2. As can be seen, the binding energy ($E_B$) has a downward trend, indicating that the total energy required for cluster binding decreases as the number of atoms increases. However, since the number of Ag atoms differs between clusters, the overall $E_B$ cannot be used to compare their stability, and the average binding energy ($E_b$) needs to be introduced. The calculated values of $E_b$ are shown in Table S5. It was found that $E_b$ decreases with the

increase of n, reaching the smallest value (indicative of the highest cluster stability) for n = 6. Therefore, the $Ag_6$ clusters were selected for the following experiments.

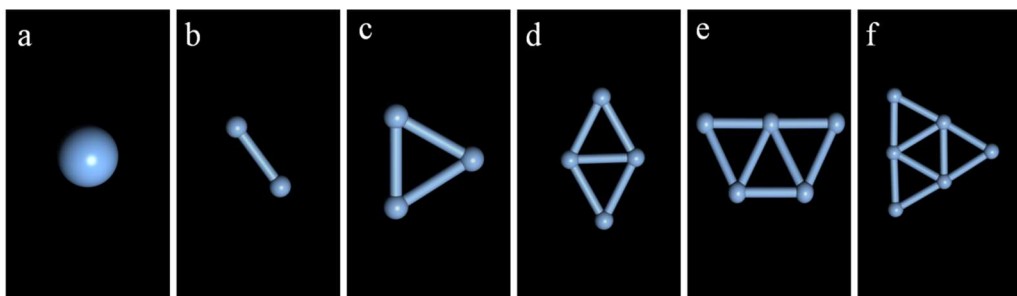

**Figure 1.** Most stable configurations of $Ag_n$ (n = 1–6) clusters: (**a**) $Ag_1$; (**b**) $Ag_2$; (**c**) $Ag_3$; (**d**) $Ag_4$; (**e**) $Ag_5$; (**f**) $Ag_6$.

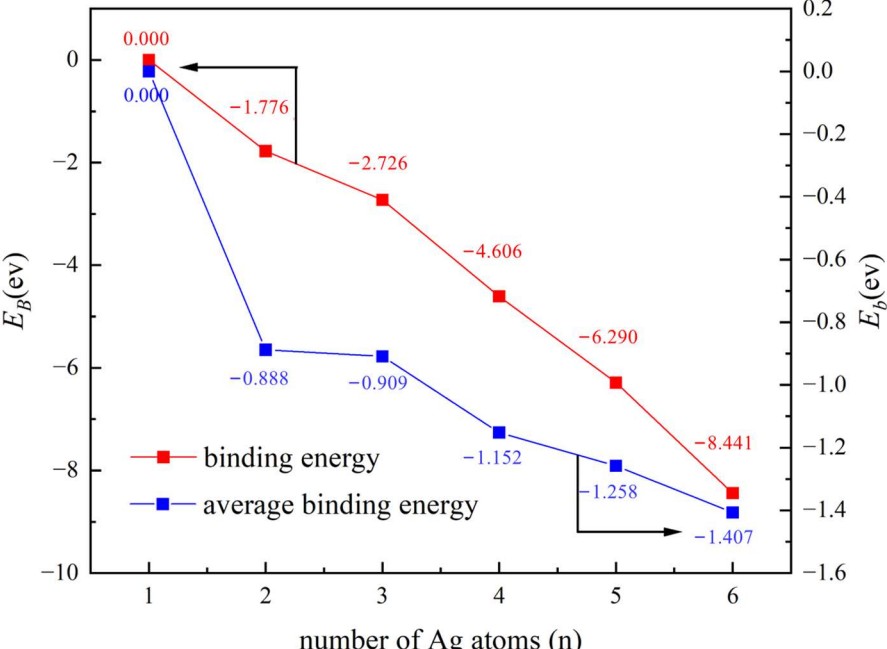

**Figure 2.** Variation of binding energy and average binding energy with the number of Ag atoms.

### 2.2. Study on the Stability of the CuAg Clusters

The selected $Ag_6$ clusters were doped with Cu atoms, and the changes in parameters such as bond length, bond angle, and stability after doping were examined. When doped with a Cu atom, the resulting $Cu_1Ag_5$ cluster has two kinds of isomers. The specific structures are shown in Figure 3. Although the two structures have symmetry, the addition of Cu changes the structure of the Ag cluster, which loses its regular triangle structure. Calculation of the total energy and average binding energy of the two structures (Table S6) revealed that structure *a* is relatively stable in comparison to structure *b*. According to the method mentioned above, the most stable configurations of the $Cu_{6-n}Ag_n$ (n = 1–5) clusters shown in Figure 4 were obtained.

The HOMO and LUMO values of the six structures in Figure 4 were simulated and calculated. Figure 5 shows that the $E_{LUMO}$ decreases after doping with Cu, indicating that the presence of the Cu atoms enhances the electron-accepting ability of the Ag clusters. Moreover, the $E_{HOMO}$ of the bimetallic clusters decreases after Cu doping, indicating that the addition of Cu atoms reduces the electron-donating ability. The ΔE value is not significantly changed by Cu doping, and the ΔE of $Cu_5Ag_1$ is the smallest, which suggests that the electron transfer is the fastest in this cluster.

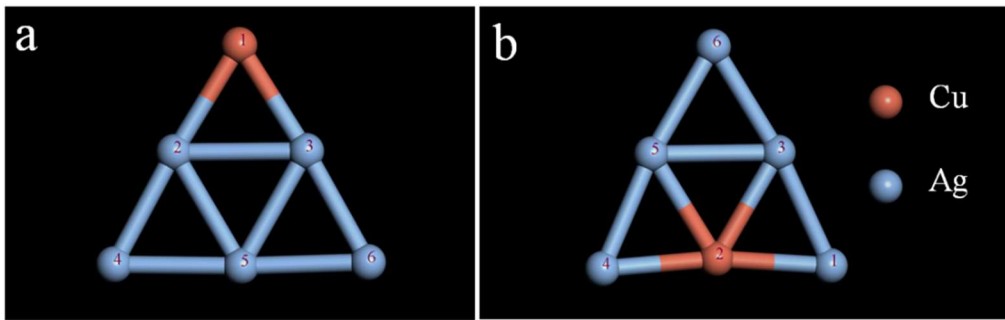

**Figure 3.** Isomeric structure of $Cu_1Ag_5$ clusters: (**a**) Cu is the tip position; (**b**) Cu is the middle position.

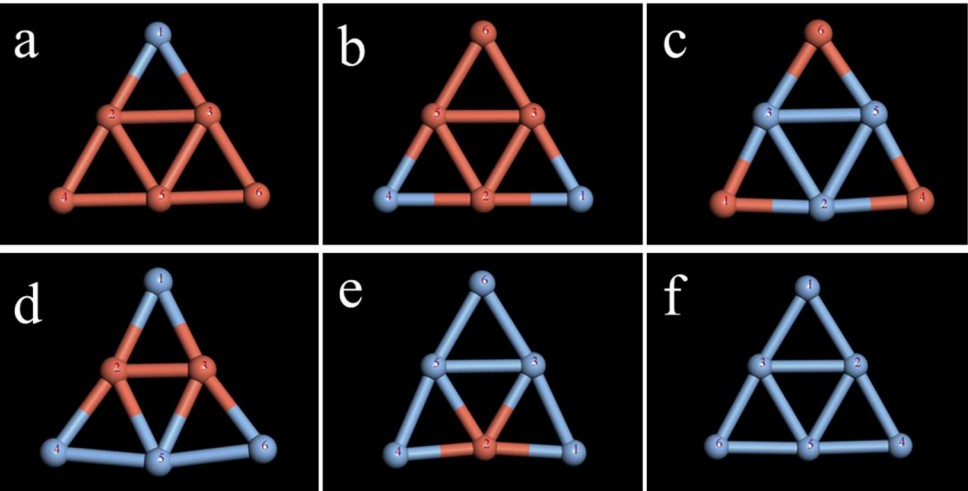

**Figure 4.** Most stable configuration of $Cu_{6-n}Ag_n$ (n = 1–6) clusters: (**a**) $Cu_5Ag_1$; (**b**) $Cu_4Ag_2$; (**c**) $Cu_3Ag_3$; (**d**) $Cu_2Ag_4$; (**e**) $Cu_1Ag_5$; (**f**) $Ag_6$.

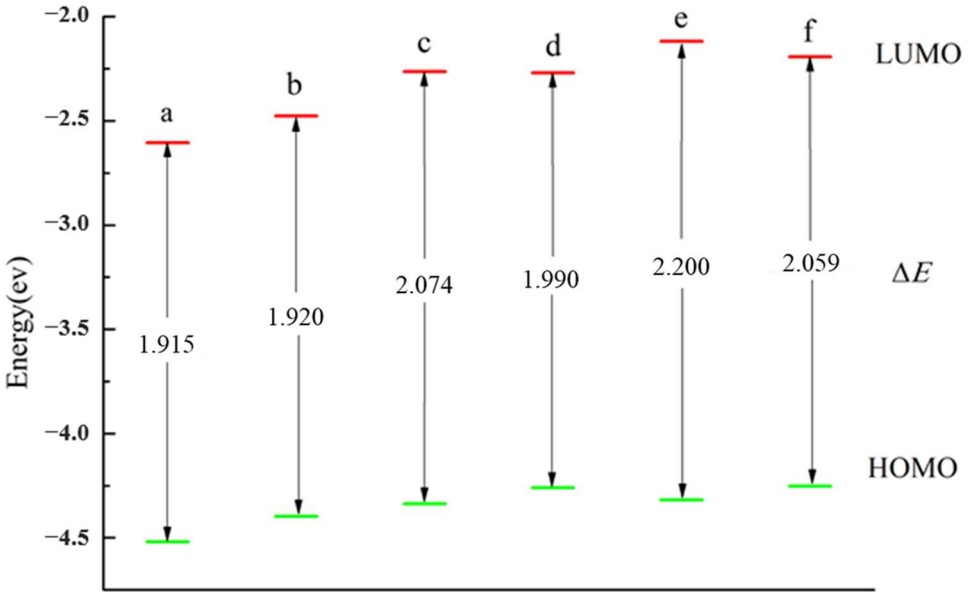

**Figure 5.** Frontier molecular orbital energy level and ΔE of the $Cu_{6-n}Ag_n$ cluster: (**a**) $Cu_5Ag_1$, (**b**) $Cu_4Ag_2$, (**c**) $Cu_3Ag_3$, (**d**) $Cu_2Ag_4$, (**e**) $Cu_1Ag_5$, and (**f**) $Ag_6$.

### 2.3. Study on the $CO_2$ Adsorption Stability of the CuAg Clusters

Table 1 lists the values of the first four atoms with the largest absolute Fukui index of the six cluster structures. Because some clusters have symmetry, the Fukui indices of atoms

that are structurally axisymmetric are equal. $CO_2$ is easily adsorbed on atoms with small Fukui (-) values because the structure is stable after adsorption; therefore, cluster catalysts with small Fukui indices would exhibit enhanced $CO_2$ activation.

**Table 1.** Fukui indices of CuAg clusters.

| Configuration | Site | Value | Site | Value | Site | Value | Site | Value |
|---|---|---|---|---|---|---|---|---|
| a | Cu (4/6) | +0.237 | Ag (1) | −0.228 | Cu (4/6) | −0.204 | Ag (1) | +0.175 |
| b | Cu (6) | +0.276 | Ag (1/4) | −0.229 | Cu (6) | −0.185 | Ag (1/4) | +0.183 |
| c | Ag (1) | +0.233 | Ag (4/6) | +0.221 | Ag (1) | −0.219 | Ag (4/6) | −0.217 |
| d | Ag (1) | +0.238 | Ag (4/6) | −0.252 | Ag (4/6) | +0.211 | Ag (1) | −0.182 |
| e | Ag (1/4) | −0.252 | Ag (6) | +0.241 | Ag (1/4) | +0.210 | Ag (6) | −0.206 |
| f | Ag (1/4/6) | −0.234 | Ag (1/4/6) | −0.231 | Ag (2/3/5) | +0.103 | Ag (2/3/5) | −0.102 |

CO$_2$ adsorption proceeds mainly by two processes, i.e., single-site adsorption and dual-site adsorption. For the $Cu_5Ag_1$ cluster, the absolute Fukui value (−) of the silver atom at position 1 is the largest, and $CO_2$ adsorption is therefore preferred at this site. The specific stable structure after adsorption is shown in Figure 6, and the bond lengths, bond angles, and energies of the $Cu_5Ag_1$–$CO_2$ structure are summarized in Table 2. In double-site adsorption, comparison of the C–O bond length before and after adsorption reveals that two C–O bonds of $CO_2$ elongate and the third shortens in structures *a* and *c*. In contrast, the two bond lengths of $CO_2$ in structure *b* remain virtually unaltered after adsorption. These results suggest that structures *a* and *c* are more conducive to inducing $CO_2$ reduction to alcohols. Then, by comparing the adsorption energy of structures *a* and *c*, it can be concluded that structure *a* is the best for double-site adsorption of $CO_2$.

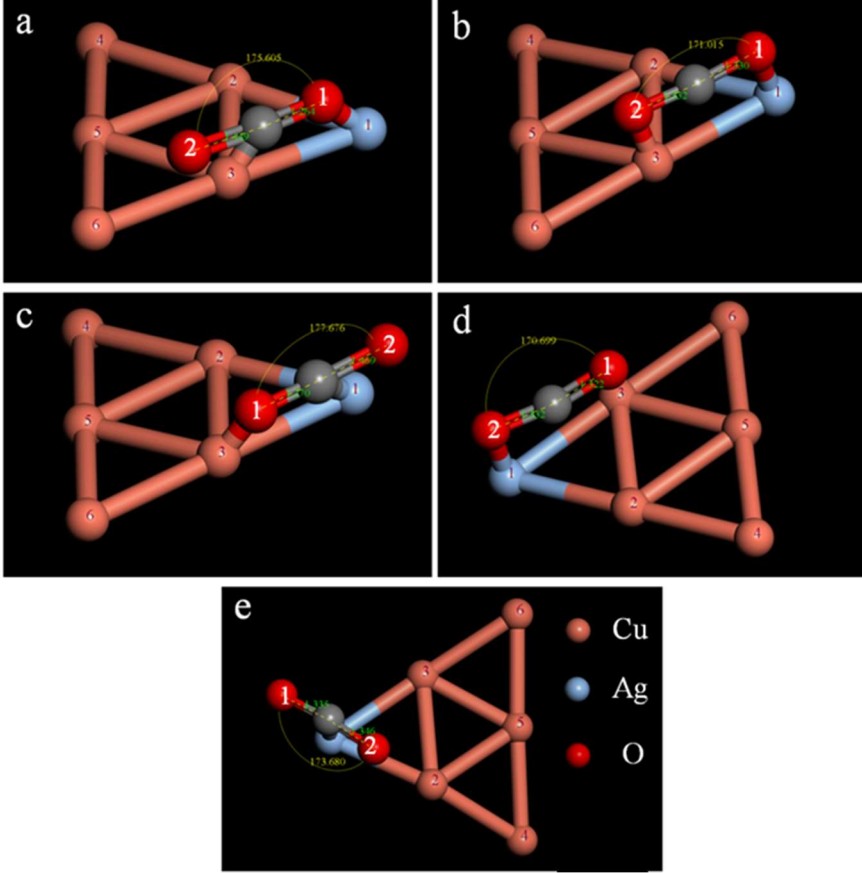

**Figure 6.** Isomers of the $Cu_5Ag_1$–$CO_2$ adsorption structure (**a**–**c**) are dual-site adsorption, while (**d**,**e**) are single-site adsorption.

**Table 2.** Bond lengths, bond angles, and energies of the $Cu_5Ag_1$–$CO_2$ structures.

| Isomers | $\angle O1CO2/°$ | $d_{C-O1}/Å$ | $d_{C-O2}/Å$ | $E_{CuAg-CO2}/Ha$ | $E_{obs}/eV$ |
|---------|------------------|--------------|--------------|-------------------|--------------|
| a | 175.605 | 1.264 | 1.449 | −13,589.920 | −0.555 |
| b | 171.015 | 1.330 | 1.332 | −13,589.911 | −0.299 |
| c | 177.676 | 1.270 | 1.469 | −13,589.905 | −0.1439 |
| d | 170.699 | 1.322 | 1.332 | −13,589.911 | −0.301 |
| e | 173.680 | 1.335 | 1.346 | −13,589.906 | −0.163 |

For single-site adsorption, the $CO_2$ bond length in structure *d* does not change significantly, whereas the change in structure *e* is more obvious, indicating a higher probability of $CO_2$ activation when C atoms are adsorbed on $Cu_5Ag_1$.

Similarly, the best adsorption configurations for double-site adsorption and single-site adsorption were obtained for the other clusters, as shown in Figures 7 and 8. The calculation results show that the adsorption method, the $CO_2$ activation degree, and the overall stability after adsorption are different between clusters.

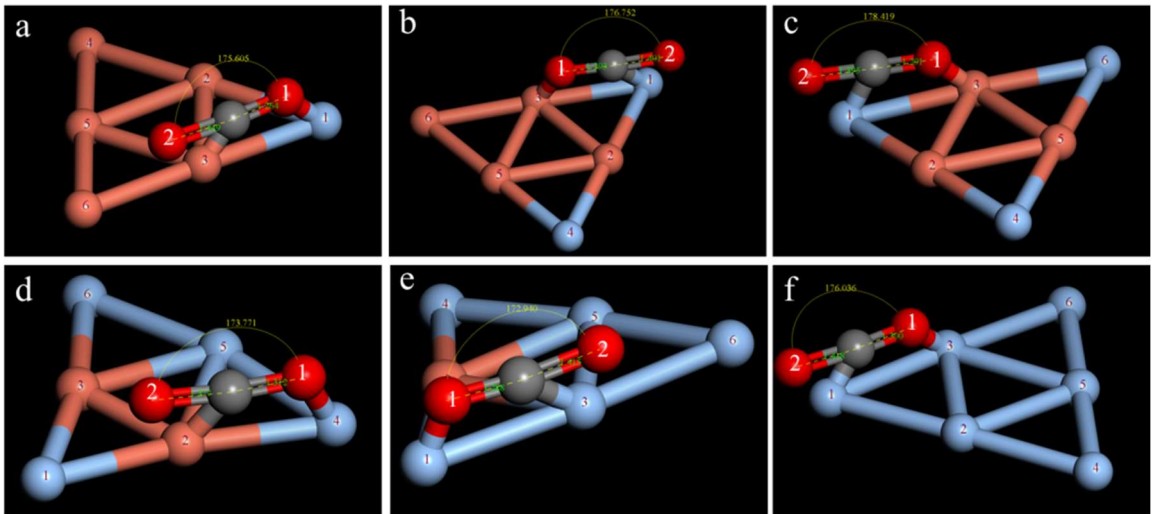

**Figure 7.** Optimal $CO_2$ adsorption structure in the $Cu_nAg_{6-n}$ cluster (n = 1–6) for dual-site adsorption: (**a**) $Cu_5Ag_1$; (**b**) $Cu_4Ag_2$; (**c**) $Cu_3Ag_3$; (**d**) $Cu_2Ag_4$; (**e**) $Cu_1Ag_5$; (**f**) $Ag_6$.

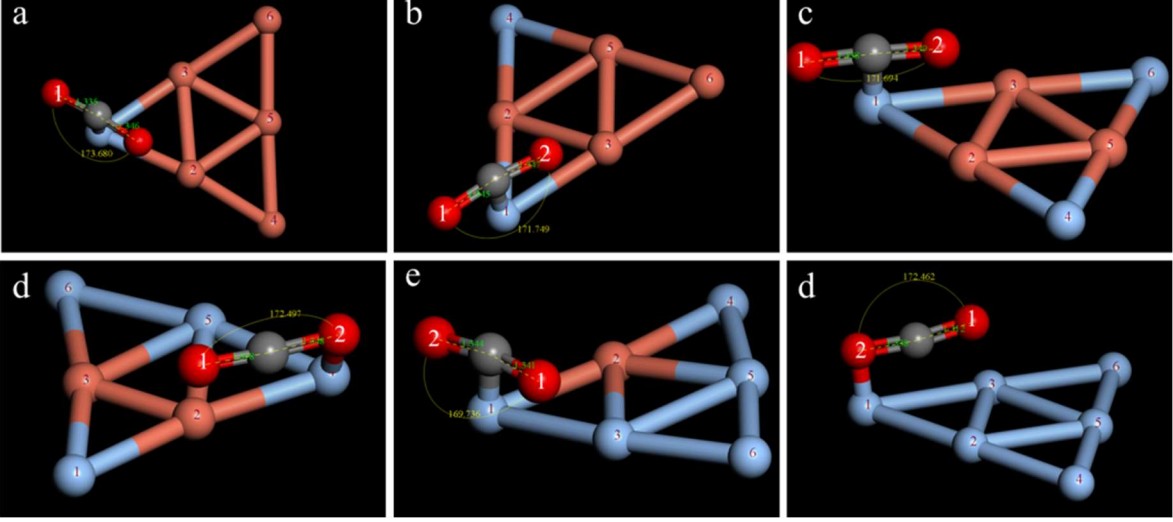

**Figure 8.** Optimal $CO_2$ adsorption structure in the $Cu_nAg_{6-n}$ cluster (n = 1–6) for single-site adsorption: (**a**) $Cu_5Ag_1$; (**b**) $Cu_4Ag_2$; (**c**) $Cu_3Ag_3$; (**d**) $Cu_2Ag_4$; (**e**) $Cu_1Ag_5$; (**f**) $Ag_6$.

### 2.4. Pathway Study on $CO_2RR$ to $CH_3OH$

The mechanism of the $CO_2$ reduction reaction is still unclear, and further experimental and theoretical research is needed. Recently, the reaction pathway of $CO_2$ hydrogenation reduction to $CH_3OH$ has been studied [40,41]. Figure 9 depicts the preferred reaction mechanism of $CO_2$ hydrogenation to obtain $CH_3OH$; thus, this route was selected to study the catalytic effect of the CuAg clusters. The specific reaction intermediates are $CO_2$*, HCOO*, HCOOH*, $H_2COOH$*, TS4, $CH_2O$*, TS5, $CH_3O$*, and $CH_3OH$.

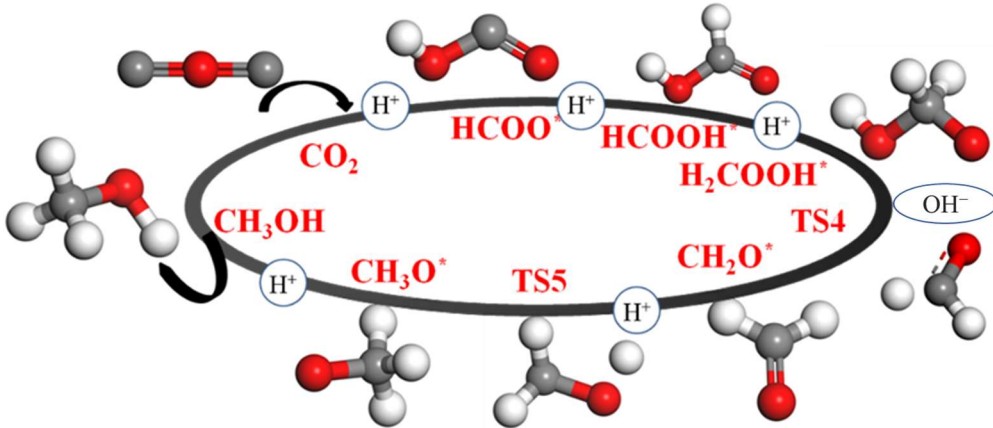

**Figure 9.** Reaction pathway of $CO_2$ hydrogenation to $CH_3OH$.

The Gibbs free energy ($\Delta G$) of the reactions involving intermediates, reactants, and products was calculated at 298.15 K, using the $CO_2$ reactant as a benchmark. The results are shown in Table S16. Similarly, the adsorption structures of all products, intermediates, reactants, and catalysts were simulated according to the previous calculation structure. Structure optimization and $\Delta G$ calculation were performed using the same method.

The calculated energy values of the $Cu_5Ag_1$ cluster are listed in Table S17, and the corresponding ladder diagram is shown in Figure 10. It can be seen that in the absence of a catalyst, the $\Delta G$ of the intermediates gradually rises, and the energy difference between $CO_2$ and $CH_3OH$ is 389.422 eV, which indicates that the reaction is not spontaneous under normal conditions.

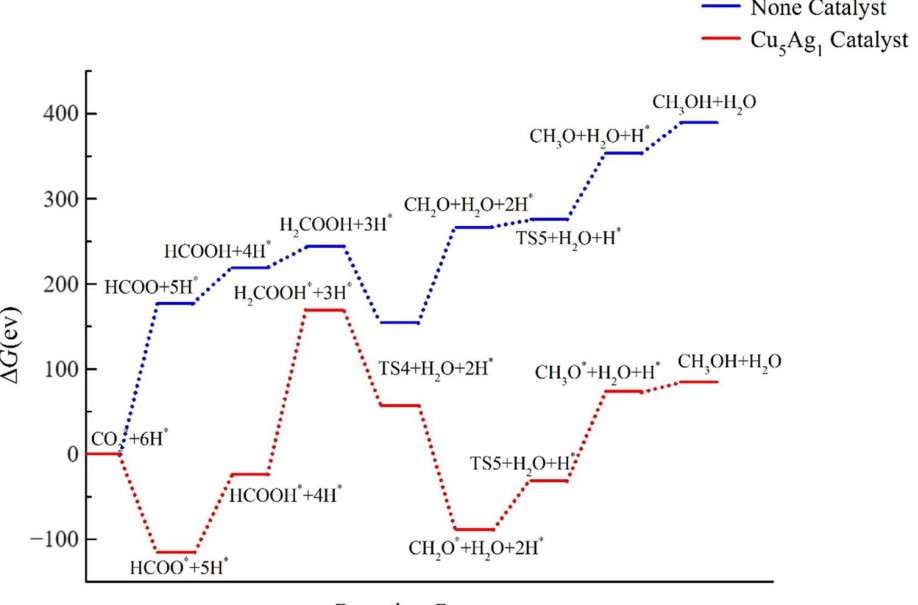

**Figure 10.** Energy diagram of $CO_2$ hydrogenation to $CH_3OH$ catalyzed by the $Cu_5Ag_1$ cluster.

When the $Cu_5Ag_1$ catalyst is added, the $\Delta G$ of the intermediate decreases compared with that obtained without the catalyst, indicating that the cluster has a certain degree of catalytic activity for the $CO_2$ reduction reaction. After adding the $Cu_5Ag_1$ cluster, the difference between the energy of the $CH_3OH$ product and that of $CO_2$ is only 84.709 eV. Compared with the results without a catalyst, although $\Delta G$ is reduced by about four times, it is still positive, indicating that the reaction cannot proceed spontaneously in the presence of $Cu_5Ag_1$. Therefore, it can be concluded that the $Cu_5Ag_1$ cluster has a catalytic effect on the reaction, albeit a small one.

The energy ladder diagram of the $Cu_4Ag_2$ cluster reduction is shown in Figure 11. Certain catalytic activity on $CO_2RR$ was detected for the $Cu_4Ag_2$ cluster by comparing the $\Delta G$ of each intermediate and the blank control data. Specifically, the catalyst decreases the $\Delta G$ of each intermediate, this decrease being the largest in HCOO* and TS4. In the absence of a catalyst, $CO_2$ has the lowest $\Delta G$ in all the pathways, but this trend changes after adding $Cu_4Ag_2$. For the $CH_3OH$ product, after adding the $Cu_4Ag_2$ catalyst, $\Delta G$ decreases by 259.869 eV. Therefore, the $Cu_4Ag_2$ catalyst can induce the formation of $CH_3OH$ to some extent, reducing the reaction energy barrier and softening the reaction conditions. However, the $\Delta G$ of $CH_3OH$ is still positive, indicating that the catalytic effect of the cluster is not ideal.

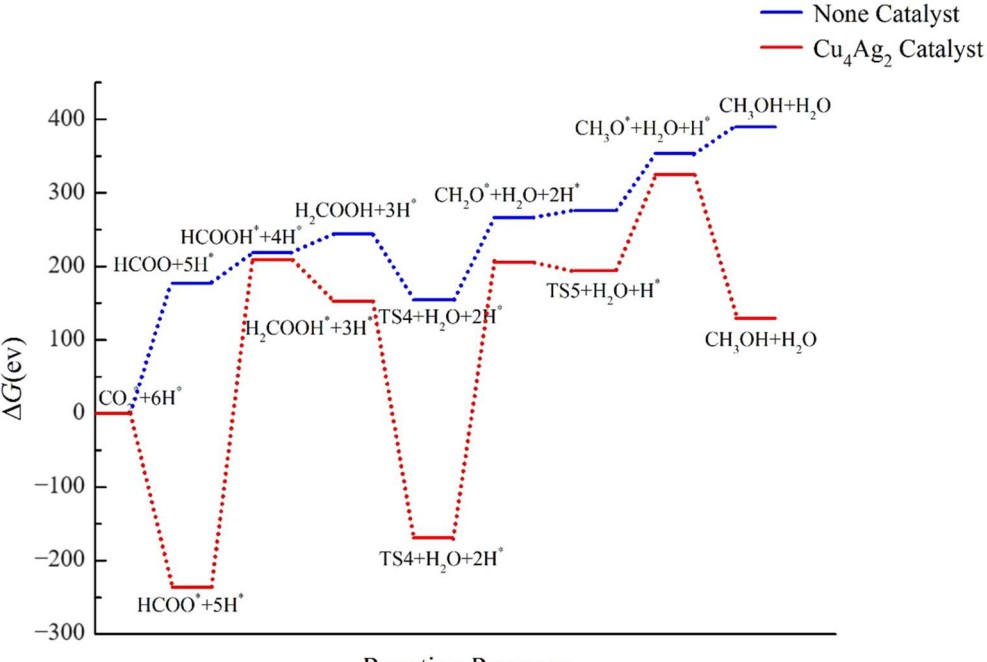

**Figure 11.** Energy diagram of $CO_2$ hydrogenation to $CH_3OH$ catalyzed by the $Cu_4Ag_2$ cluster.

The calculation results of the $Cu_3Ag_3$ cluster are shown in Figure 12. It was found that after the addition of the $Cu_3Ag_3$ cluster catalyst, the $\Delta G$ of the $CO_2$ hydrogenation route undergoes a major change. In the absence of a catalyst, the energy of the intermediates gradually rises, whereas this trend is reversed after adding the $Cu_3Ag_3$ catalyst. The catalyst reduces the $\Delta G$ of most of the intermediates in the entire path, among which HCOO*, $H_2COOH$*, and $CH_3O$* exhibit the largest decrease, indicating that the catalyst can promote the formation of the three intermediates. However, the $\Delta G$ of the transition state TS4 is higher by about 23 eV than that obtained without catalysis, which suggests that the bimetallic $Cu_3Ag_3$ cluster could not promote the formation of TS4. For the $CH_3OH$ product, after adding the $Cu_3Ag_3$ cluster catalyst, the $\Delta G$ of $CH_3OH$ decreases by 152.792 eV, indicating that the catalyst can effectively reduce the energy barrier of the reaction path. However, since the $\Delta G$ of the intermediate product increases, the catalytic effect of the $Cu_3Ag_3$ cluster on $CO_2RR$ is not significant either.

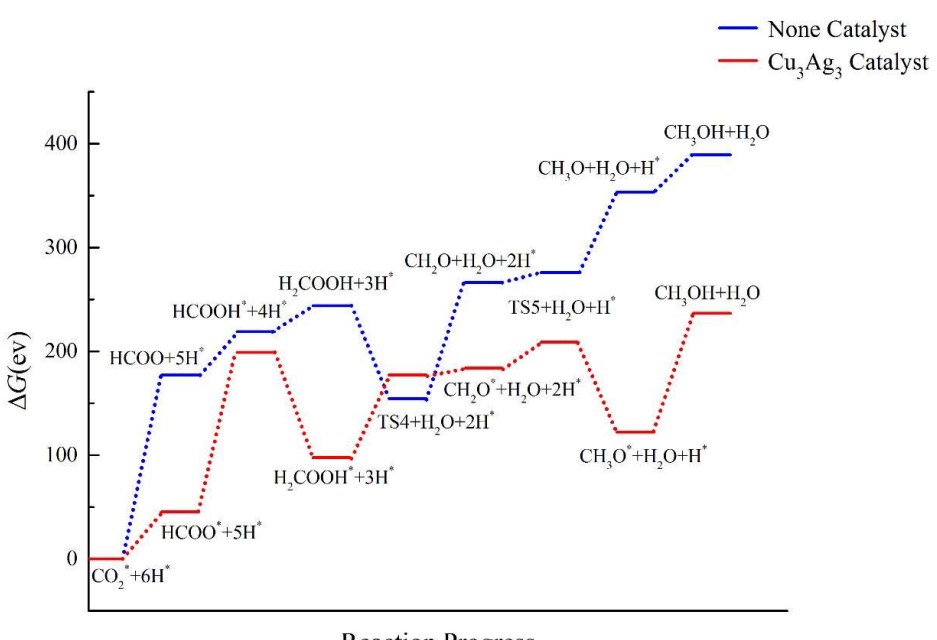

**Figure 12.** Energy diagram of $CO_2$ hydrogenation to $CH_3OH$ catalyzed by the $Cu_3Ag_3$ cluster.

Figure 13 shows the energy ladder diagram of $CO_2RR$ to $CH_3OH$ after adding the bimetallic $Cu_2Ag_4$ cluster catalyst. The $\Delta G$ of all intermediates and products decreases significantly. Compared with the results of the reaction without a catalyst, the $\Delta G$ values of all intermediates and reactants decrease directly from positive to negative, indicating that the hydrogenation of $CO_2$ to $CH_3OH$ can occur spontaneously in the presence of the $Cu_2Ag_4$ catalyst. Therefore, the $Cu_2Ag_4$ cluster displays sufficient catalytic activity on the $CO_2RR$ to $CH_3OH$.

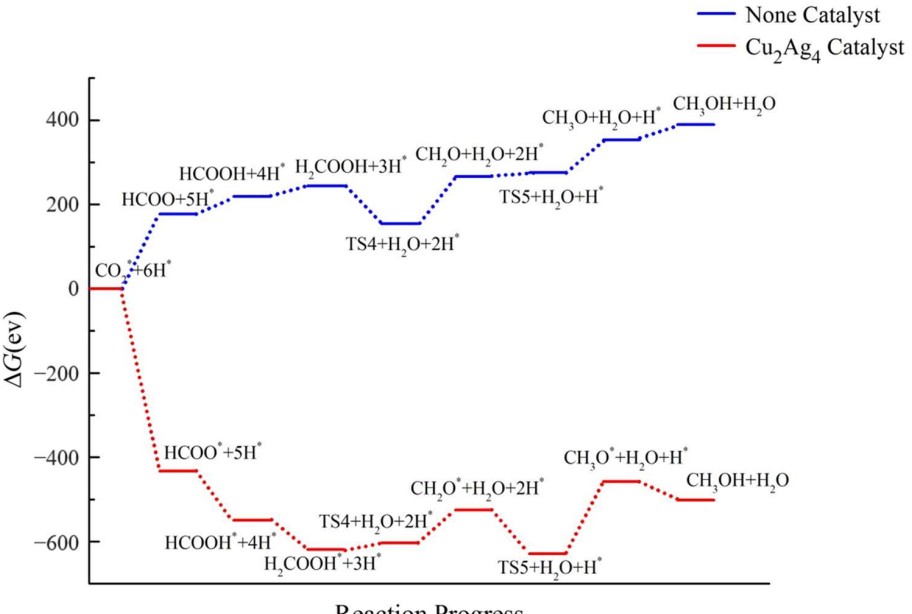

**Figure 13.** Energy diagram of $CO_2$ hydrogenation to $CH_3OH$ catalyzed by the $Cu_2Ag_4$ cluster.

The catalytic effect of the cluster doped with one Cu atom is shown in Figure 14. From the overall diagram, it can be deduced that the cluster has a good catalytic effect, because the $\Delta G$ of the six intermediates is lower than that of $CO_2$. However, the $\Delta G$ of the TS5 intermediate is higher than that of $CO_2$, and the $\Delta G$ of TS5 is positive. Focusing on the

CH$_3$OH product, it can be seen that the ΔG of CH$_3$OH after the addition of the catalyst does not differ much from that of CO$_2$; the difference is only 23.456 eV. This indicates that the reaction is more favorable after adding Cu$_1$Ag$_5$. Therefore, the Cu$_1$Ag$_5$ cluster can promote the reaction of CO$_2$ to CH$_3$OH, but the overall catalytic effect is not as significant as that of Cu$_2$Ag$_4$.

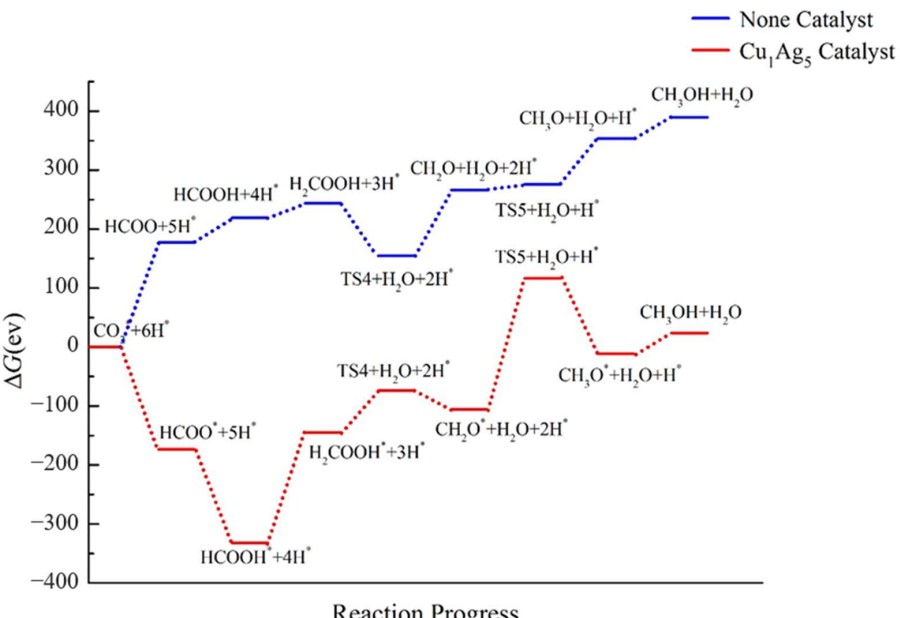

**Figure 14.** Energy diagram of CO$_2$ hydrogenation to CH$_3$OH catalyzed by the Cu$_1$Ag$_5$ cluster.

Next, we analyzed the catalytic effect of a pure silver cluster Ag$_6$. The specific calculation results are shown in Figure 15. The reaction pathway with the Ag$_6$ cluster is completely different from those of the bimetallic catalysts, and no catalytic effect on CO$_2$RR is observed. The energy barrier that needs to be overcome after adding the catalyst Ag$_6$ cluster is higher than the energy barrier present when no catalyst has been added, indicating that the pure metal Ag$_6$ cluster catalyst would require more severe reaction conditions and would destabilize the intermediates. The Ag$_6$ cluster cannot promote the formation of the intermediates in the CO$_2$RR to CH$_3$OH.

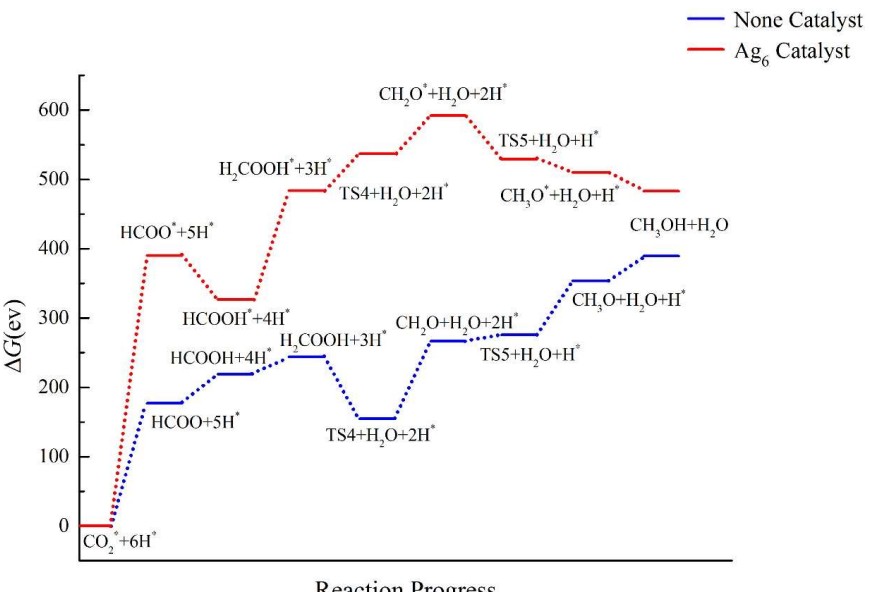

**Figure 15.** Energy diagram of CO$_2$ hydrogenation to CH$_3$OH catalyzed by the Ag$_6$ cluster.

The energy ladder diagrams of the bimetallic CuAg clusters and the pure metal $Ag_6$ cluster can be directly compared in Figure 16. It can be seen that the $\Delta G$ value of the path is higher for $Ag_6$ than for the CuAg clusters, indicating that the energy barrier that needs to be overcome for the reaction to proceed is the highest, and the catalytic effect of $Ag_6$ is the worst. This result demonstrates that Cu doping can indeed increase the catalytic activity of the clusters, which is further confirmed by the experimental results of single-site adsorption of $CO_2$ (the corresponding data can be found in the supporting information).

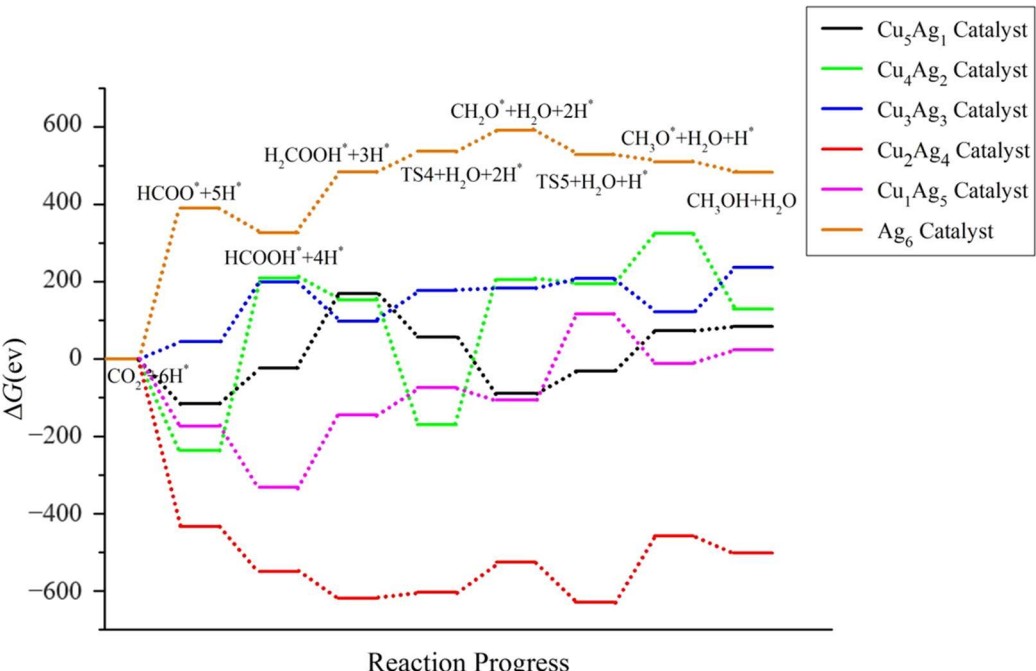

**Figure 16.** Energy diagram of $CO_2$ hydrogenation to $CH_3OH$ catalyzed by the CuAg clusters.

By comparing the catalytic activity of the doped bimetallic clusters, it was found that the $\Delta G$ of intermediates and the product obtained with $Cu_2Ag_4$ exhibits the largest decrease among the clusters, and the $\Delta G$ of all the intermediates decreases to negative values. This shows that the $Cu_2Ag_4$ catalyst can significantly improve the stability of the reaction intermediates, thereby facilitating their generation and promoting the spontaneous progress of the reaction. Therefore, among all the catalysts, $Cu_2Ag_4$ exhibits the best catalytic activity.

### 3. Models and Methods

This study primarily used the DMol$^3$ module in the Materials Studio software (version 2017), employing spin-polarized DFT to draw the structure, optimize the cluster structure, and calculate the energy. The parameters were set as follows: Functional function selects generalized gradient approximation and Perdew–Burke–Ernzerhof (PBE), the maximum iteration in energy calculation was 1000 times, and "Use symmetry" was selected. In the electronic module, the integration accuracy and self-consistent field (SCF) tolerances were both set to medium. In the basis set, double numerical plus polarization (DNP) was selected, and the basis file was set to 4.4. In the SCF module, the maximum SCF expansion was set to 1000 times, with "Use smearing to tail" checked. In the solvent module, the "Use COSMO solution environment" was set to "water".

Before designing the CuAg bimetallic cluster catalyst, the method by which it will be determined whether the catalyst itself is stable should be clarified. The binding energy *E* is calculated by quantification, and the formula is as follows (1)–(3).

### 3.1. The Cluster Stability and CO$_2$ Adsorption Structure Data Analysis

To calculate the binding energy $E_B$, find the total energy after the final step of convergence is successful in the calculation result file, and bring the total energy into Formula (1). For Ag clusters, the smaller the binding energy is, the more stable the structure is.

$$E_B = E_{Agn} - \text{n} \times E_{Ag} \tag{1}$$

The smaller the average binding energy ($E_b$) of each atom, the lower the stored energy and the more stable the structure of the whole cluster. The average binding energy of the Ag cluster is found using Formula (2), and that of the CuAg metal cluster is found using Formula (3).

$$E_b = E_{Agn}/\text{n} - E_{Ag} \tag{2}$$

$$E_b = [E_{CuAg} - \text{n} \times E_{Ag} - (6 - \text{n}) \times E_{Cu}]/6 \tag{3}$$

The adsorption energy $E_{CuAg-CO_2}$ is obtained from Formula (4). The smaller the adsorption energy, the more stable the overall structure after adsorption and the greater the change in the C–O bond length, indicating that the catalyst has a better degree of CO$_2$ activation.

$$E_{ads} = E_{CuAg\text{-}CO_2} - E_{CuAg} - E_{CO_2} \tag{4}$$

### 3.2. Data Analysis of CO$_2$ Hydrogenation Reduction Path

At 298.15 K, the Gibbs free energy of reacting all monomers is based on CO$_2$, and the respective ΔG is obtained. The two sets of ΔG data calculated with and without the catalyst were made into an energy ladder diagram to analyze the catalytic effect of CuAg clusters.

### 3.3. The HOMO, LUMO and ΔE of the Cluster

The highest occupied molecular orbital (HOMO) and lowest unoccupied molecular orbitals (LUMO) eigenvalues, the HOMO-LUMO gap, and the Fukui function are the most commonly used parameters to estimate the performance of a designed catalyst. The HOMO represents the highest occupied molecular orbital. The higher the $E_{HOMO}$, the easier it is for CuAg clusters to give electrons. The LUMO represents the lowest unoccupied molecular orbital. The lower the $E_{LUMO}$, the easier it is for CuAg clusters to obtain electrons. The magnitude of the energy band gap value ($\Delta E = E_{LUMO} - E_{HOMO}$) governs the electron transfer rate in the CuAg cluster. The smaller the $\Delta E$, the faster the electron transfer rate between the HOMO and LUMO orbitals.

### 3.4. Calculation of the Fukui Index of Atoms in the Cluster

The smaller the Fukui (−) value of an atom, the more easily the atom is attacked by a Lewis base; the larger the Fukui (+) of the atom, the more easily the atom is attacked by a Lewis acid. Because CO$_2$ is a Lewis base, the smaller the Fukui (−) value, the more stable the structure will be after CO$_2$ adsorption. The better the degree of activation of the cluster catalyst to CO$_2$, the more easily the atom becomes the best adsorption site for CO$_2$.

## 4. Conclusions

On the basis of quantum chemistry theory, the stability of pure Ag metal clusters and bimetallic CuAg clusters and the CO$_2$ adsorption activity and catalytic effect of the CuAg clusters on the CO$_2$ hydrogenation to CH$_3$OH were analyzed by energy calculation and optimization of the constructed structures. The following conclusions were obtained: The most stable configurations of Ag$_n$ (n = 1–6) metal clusters are planar. The stability of the configurations increases with the symmetry. When the adsorption site and adsorption mode of CO$_2$ on the cluster catalyst are different, the degree of activation of the catalyst and the stability after the adsorption of CO$_2$ are also different. Doping Cu atoms in the Ag cluster can improve the catalytic activity toward CO$_2$ hydrogenation to CH$_3$OH, with the Cu$_2$Ag$_4$ cluster

affording the best results. This theoretical study provides guidance and a reference for future work in the design of mixed-metal catalysts with high $CO_2RR$ performance.

Based on theoretical calculations, the relationship between a catalyst's structure, composition, and catalytic $CO_2RR$ activity is constructed, which can be used to guide the preparation of the catalyst and better study the reaction path and mechanism of $CO_2RR$ synthesis of $CH_3OH$. This has important theoretical and practical significance for promoting research and development of catalytic $CO_2RR$ synthesis of $CH_3OH$. This article establishes a database for future research and further guides the experimental study of catalysts for reducing $CO_2$ to $CH_3OH$. In future studies, theoretical research will be used in the catalytic $CO_2RR$ to produce alcohols, methane, formic acid, and formaldehyde.

**Supplementary Materials:** The following supporting information can be downloaded at: https://www.mdpi.com/article/10.3390/catal14010007/s1.

**Author Contributions:** Investigation, S.X., X.L. and Q.Z.; writing—original draft preparation, S.X., X.L., Q.Z. and A.L.; writing—review and editing, X.R., L.G. and T.M. All authors have read and agreed to the published version of the manuscript.

**Funding:** The support of the Fundamental Research Funds for the Central Universities (DUT22LK09), the Open Foundation of Key Laboratory of Industrial Ecology and Environmental Engineering, MOE (KLIEEE-20-01, KLIEEE-21-02), the Foundation of State Key Laboratory of High-efficiency Utilization of Coal and Green Chemical Engineering (2022-K70), and the Hefei Advanced Computing Center for this work are gratefully acknowledged.

**Data Availability Statement:** Data generated or analyzed during this study are provided in full within the article.

**Conflicts of Interest:** The authors declare no conflict of interest.

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
