# Peer review of "Density Functional Theory Study of CuAg Bimetal Electrocatalyst for CO2RR to Produce CH3OH"

_catalysts, doi:10.3390/catal14010007_

Round 1

Reviewer 1 Report

Comments and Suggestions for Authors

Author Response

Reviewer #1:

  1. Comment: “1. Improve the abstract with quantitave results and by including the most relevant results.”

Response to comment:

Thank you very much for your valuable comment.

I In the submitted revision, the abstract section has been improved with relevant results and quantitative results.

  1. Comment: “Improve models and methods section in such a way that the provided informa6on allows to replicate the conducted procedures.”

Response to comment:

Thank you very much for your valuable suggestion.

In the submitted revision, we have optimized the model and method sections with study details added to facilitate the repetition of the performed procedures.

  1. Comment: “The title does not correspond to the content, mainly in sense of …”electrocatalyst for CO2RR”. It should be clearly stated within your manuscript what makes of the assessed material an electrocatalyst, what is the evidence of the “highperformance” mentioned in the title. The abstract does not correspond to the main findings of the work.”

Response to comment:

Thank you very much for your valuable suggestion.

The title was revised as “Density functional theory study of CuAg bimetal electrocatalyst for CO2RR to produce CH3OH”, and the abstract was revised to reveal the study of this manuscript.

  1. Comment: “Improve discussion in sec6ons 3.1 and 3.2.”

Response to comment:

Thank you very much for your valuable comment.

In the submitted revision, we refined the discussion of 3.1 and 3.2.

  1. Comment: “L. 173 discuss why alcohols instead of methane, formic acid or formaldehyde, are expected, and also why figure 9 stops with methanol and the carbon chain stops growing.”

Response to comment:

Thank you very much for your valuable comment.

The purpose of this work is to theoretically study a series of CuAg cluster catalysts and establish a screening mechanism to calculate the catalytic activity of reducing CO2 to CH3OH. The use of computational chemistry can accurately design catalyst configurations, predict reaction activity, study CO2RR mechanism, display reaction paths and the state of various intermediates on the atomic scale, and accurately display the reaction process. Meanwhile, catalysts can be precisely designed and CO2RR performance predictions can effectively avoid interference factors leading to errors in research results. On the basis of theoretical calculations, the relationship between catalyst structure, composition and catalytic CO2RR activity is constructed, which can be used to guide the preparation of catalyst and better study the reaction path and mechanism of CO2RR synthesis of CH3OH, which has important theoretical and practical significance for promoting the research and development of catalytic CO2RR synthesis of CH3OH.

In future Different from the previous published articles, the research target products mainly focus on CO, CH4, HCOOH and C2H5OHH, while the reduction of CO2 to CH3OH has not been reported. Among them, CH3OH is one of the most important chemicals used as green fuels or intermediates of chemical reactions. It can be directly used in energy conversion systems, such as methanol fuel cells, or internal combustion engines due to its relatively high energy density. Almost all currently commercially used CH3OH is produced by syngas, and the development of carbon dioxide electrochemistry has led to new research directions in the production of CH3OH. Therefore, this article will establish a database for future research and further guide the experimental study of catalysts for reducing CO2 to CH3OH.

In future studies, the theoretical research will be used in the catalytic CO2RR to produce catalytic CO2RR to produce alcohols, methane, formic acid, and formaldehyde.

  1. Comment: “Discuss how can the Gibbs free energy be related to the current that must be applied to the electrodes.”

Response to comment:

Thank you very much for your valuable comment.

The Gibbs free energy was related to the RDS of catalytic CO2RR reactions, which can be used in the experiment for the current and potential applied to the electrodes.

The free energy of the intermediates was acquired by Eq. 1.

G=EDFT+ZPE-TS+Gu                                         Eq. 1

In which, EDFT is the total energy, ZPE represents the zero-point energy, T is the temperature (K) and S is the entropy, Gu is the influence of electrode potential. Gu can be calculated according to -neU, where n is the number of transferred electrons, and U is the applied electrode potential.

  1. Comment: “Discuss what would be the effect of an experimental environment on results, i.e.electrolyte, CO2 species in solu6on, products (i.e. formic acid, formaldehyde,methanol), applied current density.”

Response to comment:

Thank you very much for your valuable suggestion.

potential applied to the electrodes was studied following the Eq.,G=EDFT+ZPE-TS+Gu                                            

In which, EDFT is the total energy, ZPE represents the zero-point energy, T is the temperature (K) and S is the entropy, Gu is the influence of electrode potential. Gu can be calculated according to -neU, where n is the number of transferred electrons, and U is the applied electrode potential.

In the solvent module, the Use COSMO solution environment was set to water.

And the other effect of the experimental environment will be deeply discussed.

Reviewer 2 Report

Comments and Suggestions for Authors

Author Response

Comments to the Author: “The manuscript is devoted to the theoretical study of bimetallic compound as a highly efficient material for electrocatalytic reduction of CO2. The relevance of the study is undoubted, because the problem with global warming concerns the whole planet. And in the last few decades the climate has been changing noticeably and it is CO2 that makes a great contribution to this.Nevertheless, the manuscript is not without shortcomings.”

Response to comment:

Thank you very much for your review of this manuscript and important suggestions. It is valuable and very helpful for improving this article.

We have considered all of your comments carefully and have made revisions accordingly, the review manuscript was carefully organized and written.

  1. Comment: “There is no caption in Figure 2, which refers to the red and blue lines.”

Response to comment:

Thank you very much for your valuable suggestion.

In the submitted revision, we refined Figure 2 by adding annotations for the red and blue lines.

  1. Comment: “In the manuscript, a structure with 6 clusters of silver doped with copper atoms is chosen. The authors limited themselves to flat objects of study, while in reality we are faced with 3D objects. In this regard, we would like to hear the authors' arguments in favor of 2D structure.”

Response to comment:

Thank you very much for your valuable comment.

In Agn (n=1~6) metal clusters, the most stable configurations of the clusters are all planar structures when n≤6. The most stable structure is the Ag atom itself when the number of atoms n=1, Ag2 is linear, Ag3 is most stable as a planar square triangle, Ag4 is a planar rhombus, Ag5 is a planar trapezium consisting of three triangles, and Ag6's most stable conformation is a planar square triangle consisting of a stack of four small triangles. So we chose the most stable 2D Ag6 structure as the basis for Cu doping studies.

  1. Comment: “In Fig. 6, the color of the hydrogen atom does not match the color in the figure.”

Response to comment:

Thank you very much for your valuable suggestion.

In the submitted revision, we have corrected the problem of "the colour of the hydrogen atom does not match the colour in the figure" in Figure 6.

  1. Comment: “In Table 2 Bond lengths, bond angles, are given to the third digit and energies of the Cu5Ag1-CO2 structures to the 7th digit. Is there any physical sense of such accuracy? One can just as well write the accuracy up to 20 digits. It should be limited to a significant value. It is the same on 206, 211, 224, 239, 258 line with energy. The same observation applies to supplementary information..”

Response to comment:

Thank you very much for your valuable suggestion.

In the submitted revision, we changed the number of digits in the picture and in the table to be consistent to the third digit, and we also changed the same issue in the supplementary information.

  1. Comment: “It is desirable for the manuscript to discuss recent work on electrocatalysis, such as:

Choi, C., Cai, J., Lee, C. et al. Intimate atomic Cu-Ag interfaces for high CO2RR selectivity towards CH4 at low over potential. Nano Res. 14, 3497–3501 (2021).https://doi.org/10.1007/s12274-021-3639-x

Zhu, W., Tackett, B.M., Chen, J.G. et al. Bimetallic Electrocatalysts for CO2 Reduction. Top Curr Chem (Z) 376, 41 (2018).https://doi.org/10.1007/s41061-018-0220-5”

Response to comment:

Thank you very much for your valuable suggestion.

In the submitted revision, the literature above had been cited in the part of “introduction”.